# Left Heavy Tails and the Effectiveness of the Policy and Value Networks in DNN-based best-first search for Sokoban Planning

## Abstract

Despite the tremendous success of applying traditional combinatorial search methods in various NP-complete domains such as SAT and CSP as well as using deep reinforcement learning to tackle two-player games such as Go, certain classes of PSPACE-hard planning problems have remained out of reach. Even carefully designed domain-specific solvers can fail quickly due to the exponential combinatorial search space on hard instances. Recent work that combines traditional search methods, such as best-first search and Monte Carlo tree search, with Deep Neural Networks' (DNN) heuristic prediction has shown promising progress. These methods can solve a significant number of hard planning instances beyond specialized solvers. To better understand why these approaches work, we studied the interplay of the policy and value networks in DNN-based best-first search on the Sokoban domain and show the surprising effectiveness of the policy network, further enhanced by the value network, as a guiding heuristic for the search. To further understand the phenomena, we studied the cost distribution of the search algorithms and found that Sokoban planning instances can have heavy-tailed runtime distributions, with tails both on the left and right-hand sides. In particular, for the first time, we show the existence of *left heavy tails* and propose an abstract tree model that can empirically explain the appearance of these tails. We provide extensive experiment data supporting our model. The experiments show the critical role of the policy network as a powerful heuristic guiding the search, which can lead to left heavy tails with polynomial scaling by avoiding exploring exponentially sized sub-trees. Our results also demonstrate the importance of random restart strategies, as are widely used in traditional combinatorial solvers, for DNN-based search to avoid left and right heavy tails.

## 1 Introduction

Combinatorial search is a key domain for artificial intelligence. Unfortunately, combinatorial domains usually have intractable theoretical complexity, such as NP-complete, PSPACE-complete, or even undecidable. In the past few decades, we have observed tremendous progress for practical problem solving in NP-hard domains with wide applicability in, for example, circuit design (Hong et al., 2010), hardware verification (Gupta et al., 2006), and mathematical discovery (Konev & Lisitsa, 2014). SAT solvers based on conflict-driven clause learning can solve instances with thousands of variables and clauses in seconds, which demonstrates surprising scaling performance despite SAT being an NP-complete task (Silva & Sakallah, 2003).

In contrast, practical combinatorial search in PSPACE-hard domains has remained a significant practical challenge. PSPACE-hard problems can be generally divided into two main categories: *two-player games*, such as Go, Chess, and Amazons (Lichtenstein & Sipser, 1980; Fraenkel & Lichtenstein, 1981; Hearn, 2005), and *single-agent planning problems* (a.k.a. combinatorial puzzles), such as Sokoban and problems formalized by PDDL (Planning Domain Definition Language) (Culberson, 1997; Bylander, 1994). Recent achievements in the deep learning community inspired an approach of augmenting Monte Carlo tree search (MCTS) with Deep Neural Networks' (DNN) heuristic predictions. AlphaGo (Silver et al., 2016) became the first Go software to beat professional

human players in 2016, and its newer and more general version AlphaZero (Silver et al., 2017) can achieve, *tabula rasa*, superhuman performance in many other challenging game domains.

These advances naturally raise the question of whether we can build a general framework to learn different planning domains with minimum modifications. A key challenge is that hard planning instances often require intricate action sequences with hundreds of steps, and any deviation can lead to a dead-end state with no path to goal states. To address this issue, a more systematic and complete search, such as best-first search, is preferred over MCTS (Agostinelli et al., 2019). Traditional search methods augmented with neural networks' heuristics have shown promising progress in planning domains (Shen et al., 2020; Rivlin et al., 2020; Ferber et al., 2020; Feng et al., 2020a;b). These methods can solve a significant number of hard planning instances that specialized solvers cannot solve. To gain a better understanding of these approaches, we studied the interplay of the *policy* and *value networks* in DNN-based best-first search algorithms. Our experiments show the surprising effectiveness of the policy network, further enhanced by the value network, as a guiding heuristic.

To further understand the phenomena, we also studied the cost distribution of DNN-based search. Specifically, we explored and generalized the cost distribution profiles of search methods from NP-hard domains to PSPACE-hard planning domains, and show heavy-tailed cost distributions exist ubiquitously among planning instances. For the first time, we found and characterized **left heavy tails**, which are different from well-studied **right heavy tails** (also abbreviated as heavy tails in the literature). Left heavy tails occur when instances become extremely hard and most runs cannot finish in a reasonable time limit. The solver needs to be "lucky" and occasionally hit a short run. In contrast, right heavy tails characterize mildly hard instances whose majority randomized runs have short runtime. Meanwhile, the solver can be "unlucky" and occasionally hit an extremely long run which makes the expected runtime exponential. We propose an abstract search tree model that can empirically explain the appearance of left heavy tails and provide extensive experiment data supporting our model. The experiments show the critical role of the policy network as a powerful heuristic guiding the search, which can lead to left heavy tails with polynomial scaling by avoiding exploring exponentially sized sub-trees.

Randomized combinatorial solvers use various techniques, such as randomized tie-breaking and random variable ordering, to carefully inject a *controlled amount* of randomization into a deterministic search procedure (Crawford & Baker, 1994; Bresina, 1996; Gomes et al., 1998). The randomization step requires careful engineering and analysis of the solver since excess randomization can hamper the effectiveness of random restarts. In our approach, we found that *uncertainty-aware networks* (Huang et al., 2017; Chua et al., 2018; Sedlmeier et al., 2019; Malinin & Gales, 2018) can provide just the right amount of controllable randomization into a deterministic search algorithm.

In this paper, we consider Sokoban as the background planning domain. The only domain knowledge neural networks and search algorithms receive is 1) input state representation; 2) state-action transition function; 3) deciding goal states. These three components are the *minimum* requirements to describe any planning domain and we did not use any other Sokoban-specific techniques, such as dead-end detection. We evaluated DNN-based search on more than $10,000$ instances with significant variations in underlying combinatorial structure.

Our results also demonstrate the importance of *random restart strategies*, as are widely used in traditional combinatorial solvers, for deep reinforcement learning (DRL) and deep AI planning systems to avoid left and right heavy tails. In summary, our overall contributions are as follows:

1. We studied *the interplay of the policy and value networks* on DNN-based best-first search for the Sokoban domain. Our experiments show *the surprising effectiveness of the policy network, further enhanced by the value network,* as a guiding heuristic for the search.

2. We studied the runtime distribution on more than $10,000$ instances and propose distribution-independent statistics to quantify the heaviness of tails and effectiveness of random restarts. *For the first time*, we extensively studied **left heavy tails** from experiment data, introduce an *abstract search tree model* with *critical nodes*, and formally show how left heavy tails can arise during the search. Left heavy tails of runtime distributions are explained by the critical role of the policy network as a guiding heuristic. Polynomial runtime scaling can occur because the policy network helps avoid exploring exponentially sized sub-trees during the search.

3. We show the importance of using *uncertainty-aware* networks in the planning domain and how it can add a controllable amount of randomization to a deterministic solver. We show how a *restart*

*strategy* can improve DNN-based search's effectiveness. In particular, our experiments show for larger budgets, more frequent restarts are more effective.

## 2    BACKGROUND AND RELATED WORK

**Sokoban as a planning domain.**    Sokoban is a PSPACE-complete puzzle whose goal is to push a set of boxes into the same number of goal cells in a grid maze with walls (Culberson, 1997). Sokoban is among the most challenging known AI planning domains. The domain remains challenging even for specialized solvers with significant human domain knowledge (Fern et al., 2011; Junghanns & Schaeffer, 2001). Due to its general search structure and hardness, we use Sokoban as our background domain throughout the paper.

**Optimal speedup of Las Vegas algorithms.**    Let $\mathcal{A}$ be a randomized algorithm that always outputs the correct answer when it halts but whose running time is a random variable $r^{\mathcal{A}} : \mathbb{Z}_\infty^+ \to \mathbb{R}_0^+$. Luby et al. (1993) proved that when we have full knowledge about the distribution $r^{\mathcal{A}}$, the optimal strategy that achieves the minimum expected time required to obtain an output from $\mathcal{A}$ is to repeatedly run $\mathcal{A}$ for the same amount of time $t^{\mathcal{A}}$ until it halts. To calculate $t^{\mathcal{A}}$, let

$$l(t) = \frac{1}{\sum_{x \leq t} r^{\mathcal{A}}(x)} (t - \sum_{x < t} \sum_{y \leq x} r^{\mathcal{A}}(y))$$

be the expected halting time of repeatedly running $\mathcal{A}$ with time limit $t$. Define $l^{\mathcal{A}} = \inf_{t < \infty} l(t)$ and $l^{\mathcal{A}}$ is finite for any non-trivial distribution $r^{\mathcal{A}}$, i.e., $r^{\mathcal{A}}(\infty) < 1$. Let $t^{\mathcal{A}}$ be any finite value of $t$ such that $l(t) = l^{\mathcal{A}}$, if such a value exists, or $t^{\mathcal{A}} = \infty$ otherwise. Luby et al. (1993) also showed that when $r^{\mathcal{A}}$ is unknown, the universal strategy that runs $\mathcal{A}$ for time limit

$$1, 1, 2, 1, 1, 2, 4, 1, 1, 2, 1, 1, 2, 4, 8, 1, 1, 2, 1, 1, 2, 4, 1, 1, 2, 1, 1, 2, 4, 8, 16, ...[1]$$

can achieve estimated halting time $O(l^{\mathcal{A}} \log(l^{\mathcal{A}}))$ for any randomized algorithm $\mathcal{A}$. This bound is optimal among all universal strategies.

**Right heavy tails in randomized search.**    Gomes et al. (2000; 2005) observed randomized search on SAT and CSP can exhibit right heavy tails, in particular for so-called under-constrained instances, i.e., the majority of randomized runs on the same instance halt in a relatively short time while a non-negligible fraction of extremely long runs makes the average running time exponential. Gomes et al. (2000) formalized the runtime with the Pareto-Lévy form

$$P(X > x) \sim Cx^{-\alpha}, x > 0$$

and showed random restarts can dramatically reduce the runtime variance and potentially eliminate right heavy tails.

**Uncertainty-aware network.**    There is a line of research on the uncertainty of neural networks to reduce test error, provide confidence estimate, and improve model-based reinforcement learning (Huang et al., 2017; Chua et al., 2018; Sedlmeier et al., 2019; Malinin & Gales, 2018). Our method augments neural networks with Monte Carlo (MC) dropout to introduce randomization to deterministic search engines (Gal & Ghahramani, 2016). MC dropout enables dropout layers during testing and the dropout rate can control the amount of randomization. For our experiment, the uncertainty comes from two main sources: (1) the distributional mismatch between the training and test datasets; (2) noises in the training data since the remaining distances (found by specialized solvers) are usually not optimal. See Section 3 for more details about data preparation.

## 3    FORMAL FRAMEWORK

**Policy-guided best-first search.**    Best-first search is an informed search algorithm, which explores a graph by expanding the most promising node chosen according to an **evaluation function** $f(n)$ from the open set (search boundary nodes). $f(n)$ can use both the knowledge acquired so far while exploring the graph, denoted by $g(n)$, and a heuristic function $h(n)$, which estimates the remaining distance to the nearest goal state. Starting from the start state, best-first search gradually enlarges

---

[1]https://oeis.org/A182105

the current search graph by consecutively expanding a new node $n$ which minimizes the evaluation function $f(n)$ and adds $n$ to the closed set (expanded nodes). The search considers duplicate state detection and merges different nodes with the same state into a single node. Sokoban has unit cost so that $g(n)$ represents the depth of the node $n$. The heuristic function $h(n)$ is estimated by a value network, which is explained in further detail below.

Orseau & Lelis (2021) proposed Policy-guided Heuristic Search (PHS) to further learn a policy network, which takes a state $s$ as input and outputs a vector of action probabilities $p$ with components $p(a|s)$ for each valid action $a$ of $s$. Specifically, they adapted the evaluation function to

$$f(n) = \frac{g(n) + h(n)}{\pi(n)}, \pi(n) = p(s_1|s_0) \cdots p(s_m|s_{m-1}),$$

where $(s_0, ..., s_m)$ is the sequence of states from the root node to $n$. Orseau & Lelis (2021) also proposed PHS*, a variant of PHS, that uses the evaluation function $f(n) = \frac{g(n)+h(n)}{\pi(n)^{1+h(n)/g(n)}}$.

Both PHS and PHS* require computing the cumulative product of probability predictions among the whole path from the root to $n$. In this paper, we use a new evaluation function which only depends on the probability prediction of the incoming nodes of $n$:

$$f(n) = \frac{g(n) + h(n)}{\max_{x \in \text{incoming nodes of } n} p(n|x)}.$$

Experiment data shows using this simpler evaluation function can consistently solve more instances.

**Data preparation.** The study of the complexity and practical performance of search methods is greatly hampered by the difficulty in collecting real data. As an alternative, researchers heavily resort to procedurally generated instances or highly structured problem domains (Taylor & Parberry, 2011; Guez et al., 2018). The randomly generated instances lack sufficient structure and their underlying combinatorial search space is, in some sense, too regular.

Table 1: Comparison with previous DRL works on Sokoban

| Related works | avg width | avg height | avg size | avg boxes |
|---|---|---|---|---|
| I2As (Racanière et al., 2017) | 10 | 10 | 100 | 4 |
| PHS (Orseau & Lelis, 2021) | 10 | 10 | 100 | 4 |
| Feng et al. (2020a); Shoham & Elidan (2021) | 13 | 19 | 247 | 16 |
| Our setting | 12.0 | 13.6 | 183.5 | 20.2 |

To bridge this gap, we collected all the Sokoban instances from the Sokobano website[2], resulting in 10871 instances in total. All these instances were designed by different human authors in the past few decades, have great variation in the underlying structure, serve as the benchmark for specialized solvers, and also exhibit practical interest for humans to solve. The dataset is *orders of magnitude larger* than the ones used in previous works on DRL and provides a great challenge for deep heuristic learning. See Table 1 for Sokoban board statistics compared with previous works. Notice that the difficulty of Sokoban grows exponentially as the number of boxes increases.

To generate supervised training data, we ran Sokolution[3], a state-of-the-art Sokoban solver, to compute ground truth plans. Sokolution can solve 8272 out of 10871 total instances given a 10-minute time limit. We randomly divided the solved 8272 instances into a **training set** (7435 instances) and a **test set** (827 instances). For the remaining unsolved 2609 instances, we randomly sampled 200 instances and reran Sokolution with extended 2-hour time limit to solve. 137 out from 200 instances remain unsolved and we collected these 137 instances as the **hard set** to further study the cost distribution of instances that are way harder than the training instances. For each found plan $(s_0, a_1, s_1, ..., a_n, s_n)$ from the start state $s_0$ to the goal state $s_n$, we generated training tuples $(s_i, l_{s_i}, v_{s_i})$ with policy label $l_{s_i} = a_{i+1}$ and remaining distance label $v_{s_i} = n - i$ as training data.

Feng et al. (2020b) used PUSH as basic actions to achieve the state-of-the-art performance of DRL for the Sokoban domain. In this work, we use the more elementary action MOVE. A PUSH action

---

[2]http://sokobano.de/en/levels.php
[3]http://codeanalysis.fr/sokoban

can be divided into two parts: 1) moving to the correct adjacent cell for pushing a box; 2) pushing a box. As a result, PUSH requires more domain knowledge of Sokoban – the framework needs to compute all reachable cells from the current player position and decide which boxes are pushable. The number of valid PUSH actions can grow linearly on the number of boxes. In contrast, MOVE only consists of four actions (four directions), and way less domain knowledge is required to decide valid moves for any state. Using MOVE as basic actions will generate plans that are, on average, 3-4 times longer than ones generated with PUSH actions. The majority of instances considered in this paper have plans containing hundreds or thousands of moves, which provides a great challenge for AI planning.

**Network architecture and training details.** For each board state of height $H$ and width $W$, we create an input tensor of shape $[4 \times H \times W]$ with four multi-hot feature maps encoding the player position, box positions, goal cells, and player reachable cells (ignoring all boxes), respectively. The policy head outputs a vector of length four representing the probability distribution among four moving directions. The value head outputs a single scalar representing the logarithm of the estimated remaining distance. The network consists of multiple convolutional residual blocks and each block has two extra dropout layers to introduce randomization. See Appendix B for more details.

The parameters $\theta$ of the deep neural network are trained with the following loss function:

$$(p, \log(h)) = \mathrm{DNN}_\theta(s), \quad \mathrm{loss} = (\log(h) - \log(v_s))^2 - \log(p_{l_s}) + c\|\theta\|^2,$$

where $c$ is the weight decay parameter controlling $L_2$ weight regularisation.

## 4 THE INTERPLAY OF THE POLICY AND VALUE NETWORKS

Table 2: Solver statistics of solved instances on the training and hard datasets (time limit: 10 mins).

|  | Training dataset | | | | Hard dataset | | |
|---|---|---|---|---|---|---|---|
|  | expanded | time | solved | nodes per sec | expanded | time | solved |
| Sokolution | 191436 | 310 s | 100% | 618 | — | — | 0% |
| DNN-based A* | 13776 | 370 s | 93% | 37 | 18651 | 537 | **69%** |

Table 3: Solving ratio on the test dataset with various evaluation function $f(n)$ of best-first search, depending on depth $d(n)$ (a.k.a. $g(n)$), estimated remaining distance $h(n)$, estimated action probability $p(a|s)$, and cumulative path probability $\pi(n)$. Columns represent different search budget.

| Method | $f$ | Number of total node expansions (CPU runtime below) | | | | | |
|---|---|---|---|---|---|---|---|
|  |  | 1K | 2K | 4K | 8K | 16K | 32K |
|  |  | 0.5 m | 0.9 m | 1.8 m | 3.8 m | 7.4 m | 14.5 m |
| **No Policy** | | | | | | | |
| Breadth first | $d$ | 0.32% | 1.28% | 1.92% | 3.21% | 6.73% | 11.2% |
| Greedy | $h$ | 4.17% | 8.01% | 12.5% | 15.1% | 19.2% | 19.6% |
| A* | $d + h$ | 5.77% | 10.3% | 13.5% | 19.9% | 21.5% | 25% |
| WA* | $d + 2.0 \cdot h$ | 6.09% | 8.97% | 14.7% | 17.6% | 19.6% | 22.8% |
| **With Policy** | | | | | | | |
| *Pure Policy (ours)* | $1/p$ | *28.2%* | *32.1%* | *36.9%* | *40.4%* | *42.9%* | *44.6%* |
| PHS | $(d + h)/\pi$ | 15.7% | 19.9% | 23.7% | 26.3% | 29.2% | 31.4% |
| PHS* | $(d + h)/\pi^{1+h/d}$ | 28.5% | 31.4% | 38.5% | 40.1% | 44.9% | 46.2% |
| $p$ + Greedy (ours) | $h/p$ | 31.7% | 32.4% | 37.5% | 38.8% | 41.0% | 41.3% |
| $p$ + A* (ours) | $(d + h)/p$ | **32.4%** | **34.3%** | 38.8% | **43.3%** | **46.2%** | **50.0%** |
| $p$ + WA* (ours) | $(d + 2.0 \cdot h)/p$ | 31.7% | 34.0% | **39.4%** | 42.0% | 45.8% | 48.1% |

**Solver statistics.** Table 2 above shows solver statistics of Sokolution, a state-of-the-art solver of Sokoban, and DNN-based A*. We use 8 cores of Xeon 6154 CPUs for profiling both solvers (neural networks run on the CPU mode for a fair comparison). Because of the cost of evaluating the deep net search guidance, DNN-based A* expands significantly fewer nodes per second than Sokolution (about a factor of 17). Nevertheless, *given 10 minutes, DNN-based A* can solve 69% hard instances*

*that Sokolution cannot solve even given a 2-hour time limit.* So, the trained deep net provides much superior search guidance than the hand-crafted guidance in Sokolution.

**Effectiveness of policy and value networks.** Table 3 shows experiment results for different choices of the evaluation function. As shown in the table, *the policy heuristic has a significantly larger impact than the value heuristic.* Specifically, the table shows that even the Pure Policy (using only the $1/p$ term, i.e., inversely proportional to the policy prediction) significantly boosts performance compared to all value heuristics-based search strategies without the policy guidance. (See the rows above "Pure Policy" in Table 3.) With extra properly added depth and value terms, the performance of Pure Policy can further increase to obtain our best strategies with Policy + A* and Policy + Weighted A* (WA*).

To further study why the policy network is more effective, we studied the performance of both networks in detecting dead-end states since dead-ends are one crucial factor that leads to exponential runtime. In particular, we randomly sampled 2000 board states from each dataset. For each state, we used Sokolution to detect dead-end successor states. The policy/value network is considered to successfully detect a state if it predicts a higher/lower policy/value to the child on the ground truth plan than any other dead-end child. Table 4 shows that the policy network significantly outperforms the value network at detecting dead-ends, and thus can provide better search guidance.

Table 4: Dead-end detection accuracy

|        | Train | Test | Hard |
|--------|-------|------|------|
| Policy | **93%** | **81%** | **68%** |
| Value  | 41%   | 38%  | 37%  |

## 5 ANALYSIS OF LEFT HEAVY TAILS

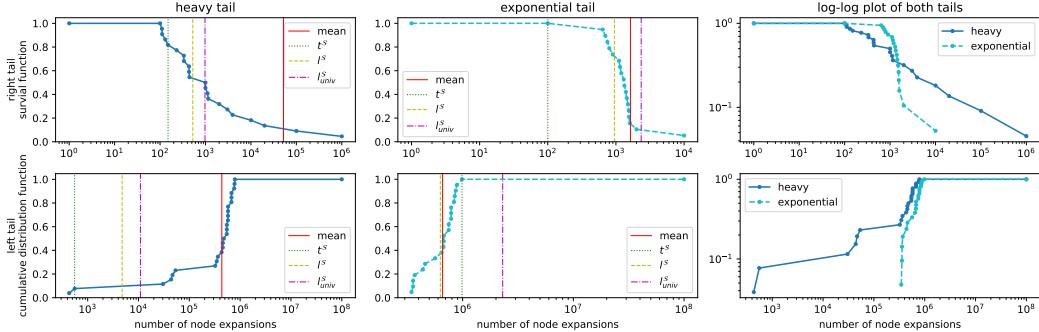

Figure 1: Each subplot shows DNN-based best-first search runtime statistics with MC dropout augmented on Sokoban instances. Each curve represents multiple runs on the same instance (instances might differ for different curves). We compare the runtime sample mean, optimal sample restart time $t^S$, expected sample total runtime with restart $l^S$, and expected total runtime of the universal strategy $l^S_{univ}$ as defined in Paragraph 5.

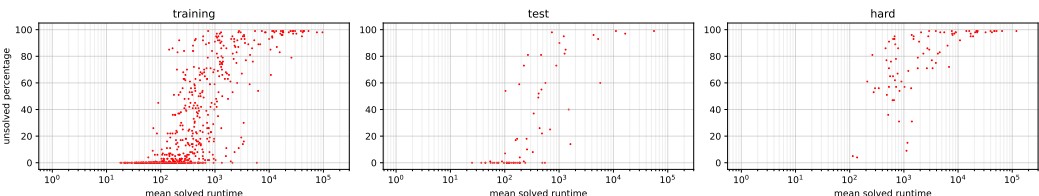

Figure 2: Subplots show randomized DNN-based best-first search results for the training, test, and hard datasets. Each red dot represents a single instance. We perform 200 randomized searches on each instance with a maximum of $300,000$ node expansions. The X-axis is the average runtime of all solved runs and the Y-axis is the unsolved percentage. Purple areas represent instances with left heavy tails while cyan areas represent right heavy tails.

**Sample statistics about the heaviness of tails.** Luby's optimal restart strategy requires the full knowledge about the runtime distribution $r^{\mathcal{A}}$ of a randomized algorithm $\mathcal{A}$. Here we introduce sample statistics that can approximate the theoretical optimal values. Specifically, for each planning instance, we perform multiple randomized searches on it with maximum search budget $\mathcal{M}$ and collect a runtime sample $\mathcal{S}$. The runtime of failed searches is recorded as $\mathcal{M}$. We approximate the optimal sample restart time $t^{\mathcal{S}}$ and expected sample total runtime with restart $l^{\mathcal{S}}$ as

$$t^{\mathcal{S}} = \arg\min_{u \in \mathcal{S}} \frac{u \cdot |\mathcal{S}|}{|\{v|v \in \mathcal{S} \text{ and } v \leq u\}|}, \quad l^{\mathcal{S}} = \frac{t^{\mathcal{S}} \cdot |\mathcal{S}|}{|\{v|v \in \mathcal{S} \text{ and } v \leq t^{\mathcal{S}}\}|}.$$

Let $T = (1, 1, 2, 1, 1, 2, 4...)$ be the time limit sequence of the universal strategy. To approximate the expected total runtime $l^{\mathcal{S}}_{univ}$ of the universal strategy on $\mathcal{S}$, let $a_i$ be the expected total runtime of applying $(T_i, T_{i+1}, T_{i+2}, ...)$ on $\mathcal{S}$ and we have the constraint

$$a_i = T_i + \frac{a_{i+1} \cdot |\mathcal{S}|}{|\{v|v \in \mathcal{S} \text{ and } v > T_i\}|}, \quad l^{\mathcal{S}}_{univ} = a_1.$$

We only need to calculate $a_i$ until the first $i$ such that $T_i \geq \mathcal{M}$ and set the remaining $a_i$ to zeros.

Figure 1 shows runtime statistics for different types of tails. Both left and right heavy tails exhibit orders of magnitude reduction of $l^{\mathcal{S}}$ and $l^{\mathcal{S}}_{univ}$ over the runtime sample mean, which demonstrates the benefit of using random restarts. For exponential tails, the expected sample total runtime with restart is very close to the sample mean, and the universal strategy even has a negative effect.

To separate the two types of heavy tails, we compared the average runtime of solved instances v.s. the unsolved percentage as shown in Figure 2. An instance is viewed as a heavy-tailed run if random restarts can reduce the expected runtime. We add further restrictions that a right/left heavy tail requires the unsolved ratio to be less/greater than 10%/90%. For experiment budget concerns, we only plotted 10% randomly sampled instances from the training and test datasets. Figure 2 shows the training dataset has almost the same number of left and right heavy-tailed instances, with the majority of instances not showing the heavy-tailed behavior. For the hard dataset, the percentage of right heavy tails decreases significantly, with more instances shifting to the top side of the figure and entering the left heavy-tailed area. We hypothesize left heavy tails occur more frequently when the underlying combinatorial structure becomes harder. Though random restarts can potentially eliminate heavy tails on both sides, left heavy tails provide further intuitions and implications for curriculum learning for DRL. In particular, they can benefit from a distributed solving procedure in which any solution found by one of the processes can be shared and learned by the curriculum framework (Weng, 2020; Narvekar & Stone, 2018; Narvekar et al., 2020).

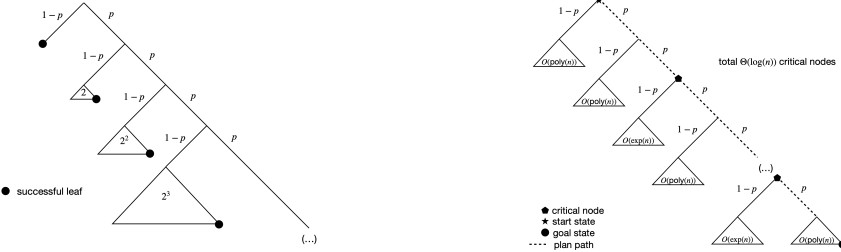

Figure 3: Left panel: imbalanced tree model for right heavy tails. Right panel: our proposed model for left heavy tails. In the left model, $p$ is the constant probability of missing a backdoor. In the right model, $p$ is the constant probability of picking the right action to the goal.

**Abstract search tree model.** Chen et al. (2001) proposed an imbalanced tree model to empirically explain right heavy tails. Here we propose an abstract tree model to empirically characterize left heavy tails. See Figure 3 for the description for both models.

Search models for planning problems differ from ones for SAT and CSP. For NP domains, the number of unassigned variables is fixed with $O(n)$ where $n$ is the problem size, and the search can assign these variables in any order. For planning domains, the search needs to assign actions in order from the start state to a goal state with potentially maximum exponential length. Our proposed model hypothesizes the existence of $O(log(n))$ **critical nodes** from which a wrong child node selected by

the search will result in extra exponential search space. As shown in the following results, our model does not depend on the actual choice of $p$ as long as $p$ is a constant value in range $(0, 1)$.

**Theorem 5.1 (The abstract tree model has exponential runtime almost surely).** *When restricting plans to polynomial length on the input size $n$, the probability that the abstract tree model has exponential runtime converges to 1 as $n$ goes infinite.* The proof is deferred to Appendix A.1.

**Theorem 5.2 (Restart achieves polynomial expected halting time).** *The optimal expected halting time with restart $l^{\mathcal{A}}$ and expected halting time using the universal strategy $l_{univ}^{\mathcal{A}}$ are both $O(poly(n))$.* The proof is deferred to Appendix A.2.

Theorem 5.1 states the runtime of the model is exponential on the input size $n$ almost surely, and Theorem 5.2 shows that the cost distribution has polynomial estimated runtime with restarts. These two theorems combined show the occurrence of left heavy tails.

Here is an intuitive explanation of the abstract tree model. For the majority of the nodes on the plan, the deep neural network either provides an accuracy heuristic to choose the right child node or makes a small error of preferring a wrong child node. As long as the error is small the search can recover from it with extra polynomial steps since the evaluation function penalizes deeper nodes. For the $O(\log(n))$ critical nodes, the error of estimated heuristics is so large that exponential search is required to jump out from the local search space.

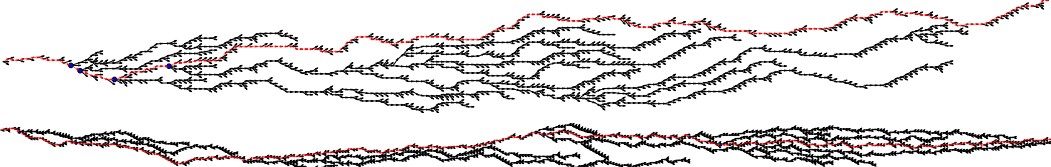

Figure 4: Two typical DNN-based best-first search graphs. We remove graph edges to form a spanning tree for clearer illustration. The search graph is built from the left to the right. Red dashed path is the plan leading form the start state to one goal state, while blue circles are *critical nodes*, which have exponential sub-search tree underneath without a near goal state.

**Structure of real search graphs.** To empirically examine our abstract model, we plotted two typical search graphs generated by DNN-based best-first search as shown in Figure 4. We found that the majority of nodes actually **do not** incur search. Networks' heuristics either directly lead the search to the correct child node, or only a very small wrong sub-tree (less than 5 nodes) is explored. *Critical nodes* (labeled as blue circles) are extremely rare in the search graph. However, when encountering such critical nodes, the search explores a large sub-tree with no near goal.

Whether AI planning systems can find a **macro action routine**, i.e., a sequence of algorithmic actions to perform a sub-goal, has a great interest for researchers. To make a long plan, e.g., prove a hard mathematical theorem, humans usually only make some *critical choices* of lemmas and schemes, and fill the remaining parts of the proof with little reasoning. Indeed, the number of required *crucial intermediate lemmas* to prove a mathematical theorem is quite small, even for challenging open problems. However, extensive and profound reasoning, search, and enumeration are needed to find such lemmas. The small number of critical lemmas compared with the long proof length reflects the prototype of such search graphs. In our experiment setting, we use MOVE as basic actions. To perform a real PUSH, the search algorithm needs to compute all reachable cells from the player's position and calculate the shortest path to the cell adjacent to the box to push. Such a long sequence of moves before performing an actual push can be viewed as a **routine**. As shown in Figure 4, the algorithm can perform a long chain of moves with little local search, which might suggest that macro action routines are **implicitly** learned as a part of neural networks' heuristics.

**Relation to backdoors.** The proposed model has a close relation to backdoors to typical case complexity. To explain why solvers scale so well in areas such as planning and finite model-checking, Williams et al. (2003) examined various benchmarks and identified that for most practically solvable problem instances, after assigning values to *logarithmic* variables, the remaining problem instance quickly becomes polynomially solvable by propagating constraints. This result illuminates

the prototypical patterns of the structure causing the empirical behavior observed in the International Planning Competitions benchmarks (Vallati et al., 2015; Cohen & Beck, 2018; Meier et al., 2014).

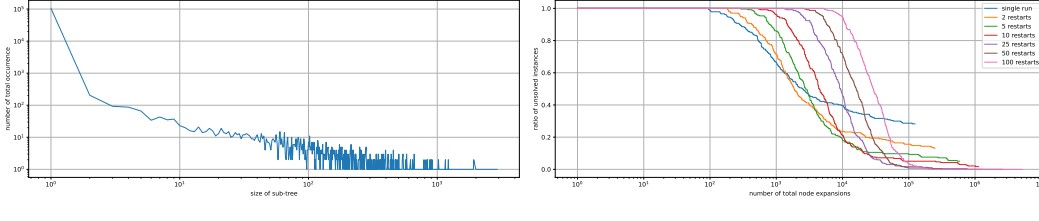

(a) Subtree Size                                      (b) Unsolved ratio

Figure 5: (a) The total number of sub-trees of 300 randomly sampled instances v.s. the tree size (in log-log scale). (b) Unsolved fraction of instances with or without restarts. Given a fixed amount of search budget (number of nodes allowed to expand), $n$-restart means to evenly divide the search budget into $n$ individual runs and union their solved instance sets as the final result.

**More experiment data about the abstract tree model.** To better illustrate the occurrence probability of different sizes of sub-search space, we randomly sampled 300 solved instances from the test dataset and counted the number of dead sub-trees (no near goal exists) for each size. As shown in Figure 5a, as the size of sub-trees grows exponentially, the number of sub-trees also decreases exponentially fast. The experiment data empirically confirms our abstract tree model that critical nodes which incur extra exponential search space have logarithmic occurrence. This is evidence about how left heavy tails occur in practical planning solving.

Hoffmann et al. (2006) explained SATPLAN, classical planning as SAT, with backdoor models. In particular, after finding and assigning logarithmic variables, the remaining problem solving becomes polynomial. In contrast, finding these critical nodes is not necessary for expand-style search algorithms. Left heavy tails are not only caused by the underlying structure of practical instances but also affected by the mysterious generalization ability of neural networks. It is an interesting future research direction to understand the surprising scaling performance of various heuristics, from conflict-driven clause learning for SAT solving to DNN-based search methods.

**Solving more instances with random restarts.** The theory of heavy-tailed cost distribution suggests that a sequence of short runs instead of a single long run may make better use of a fixed amount of computational budget. We explored this idea by considering a fixed number of total expanded nodes allowed for the search. Figure 5b shows the probability of *not* solving the instances for the test dataset. The failure rate of a single run drops the fastest for a small amount of computational budget. With more total expanded nodes, random restarts gradually achieve better performance. Specifically, to solve more instances, the solver needs to increase the total number of compute cycles. When doing so, the figure shows there comes a point where more frequent restarts are more effective. For example, with a budget of around 2,000 nodes, the strategy with 2 restarts becomes more effective than no restarts. At around 5,000 nodes, 5-restart becomes more effective than 2-restart. So, to solve a larger fraction of hard instances, more frequent restarts become more effective.

## 6 CONCLUSION

We studied the use of policy (action selection) and value (remaining distance estimate) functions as well as randomization methods for solving hard planning instances using best-first search. Our experiments show the remarkable effectiveness of the policy network and random restarts for the search. The value network provides additional global search guidance.

We show that *uncertainty-aware* networks provide an effective way to introduce randomization into the search process leading to increased efficiency. Our runtime distribution results show *heavy-tailed distributions* with tails on both the left and right-hand sides. Left heavy tails have not been observed in combinatorial search before. We also introduce an abstract computational model that explains left heavy tails. Finally, we show how *random restarts* can improve the overall search effectiveness. With larger search budgets, restarts are increasingly effective.

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

# A  THE IMBALANCED TREE MODEL PROOFS

## A.1  PROOF OF THEOREM 5.1

**Lemma A.1** *The abstract tree mode has polynomial runtime if and only if all critical nodes pick the right child.*

Proof of Lemma A.1.  Necessity comes from the structure of the model and any picking of left child will immediately result in exponential search space. For sufficiency, by the assumption, the depth of the tree model is bounded by $O(\text{poly}(n))$, and the final sub-search space with all right child picking is also $O(\text{poly}(n))$. Then we have the total runtime is $O(\text{poly}(n)) + O(\text{poly}(n)) = O(\text{poly}(n))$. Q.E.D.

Proof of Theorem 5.1.  By Lemma A.1, we have

$$
\begin{aligned}
\lim_{n \to \infty} \Pr(\text{model has poly runtime}) &= \lim_{n \to \infty} p^{\Theta(\log(\text{poly}(n)))} \\
&\leq \lim_{n \to \infty} p^{C \cdot \log(\text{poly}(n))} \qquad \text{(for some constant } C) \\
&= 0 \qquad\qquad\qquad\quad (0 < p < 1 \text{ as assumption}).
\end{aligned}
$$

As a conclusion, $\lim_{n \to \infty} \Pr(\text{tree mode has exponential runtime}) = 1$.          Q.E.D.

## A.2  PROOF OF THEOREM 5.2

Proof.  We can assume there exists at least one non-critical node on the plan (if all tree nodes are critical, we can augment an extra non-critical node to the right child of the deepest critical node without affect other properties of the tree model). Let $u$ be the shallowest non-critical node and $d$ be the depth of node $u$. We set the restart time $t^{\mathcal{A}}$ to be the size of the left sub-tree of $d$. $t^{\mathcal{A}}$ is $\text{poly}(n)$ by the definition of the model. Let $q = p^d(1 - p)$. So the expected runtime

$$
\begin{aligned}
\lim_{n \to \infty} l^{\mathcal{A}} &= \lim_{n \to \infty} (d + O(\text{poly}(n))) \sum_{k=0}^{\infty} q \cdot (1 - q)^k \cdot (k + 1) \\
&= \lim_{n \to \infty} (d + O(\text{poly}(n))) \cdot \frac{1}{q} \\
&= O(\text{poly}(n)).
\end{aligned}
$$

By the result of Luby et al. (1993), $l^{\mathcal{A}}_{univ} = O(l^{\mathcal{A}} \log(l^{\mathcal{A}})) = O(\text{poly}(n))$.          Q.E.D.

# B  NETWORK ARCHITECTURE AND TRAINING DETAILS

## B.1  NETWORK ARCHITECTURE

A single input tensor of board states has shape $[4 \times H \times W]$ and a batch of board states can have different heights and widths. For each batch, we take the maximum $H$ and $W$ of all state tensors as the batch height and width, and zero-pad the empty cells.

The network consists of a single convolution block followed by 16 residual blocks.

The convolutional block applies the following modules:

1. A convolution of 128 filters of kernel size $3 \times 3$ with stride 1
2. 2D batch normalization (Ioffe & Szegedy, 2015)
3. A ReLU nonlinearity

Each residual block applies the following modules sequentially to its input:

1. A channel-wise dropout layer with probability 30% of a channel to be zeroed.
2. A convolution of 128 filters of kernel size $3 \times 3$ with stride 1

3. 2D batch normalization

4. A Relu nonlinearity

5. A channel-wise dropout layer with probability 30% of a channel to be zeroed.

6. A convolution of 128 filters of kernel size $3 \times 3$ with stride 1

7. 2D batch normalization

8. A skip connection that adds the input to the block

9. A Relu nonlinearity

The output of the residual tower is then fed into two independent heads for computing the policy and value. Both heads contain an extra residual block followed by a fully connected layer. The policy head outputs a vector of size 4 and the value head outputs a single scalar.

## B.2 TRAINING DETAILS

We use the AdamW optimizer (Loshchilov & Hutter, 2017) with weight decay 0.01 and an initial learning rate 0.001. We train the network with supervised training data for 200 epochs. The last 50 epochs use a learning rate of 0.0001. We set batch-size to 256. The whole training procedure took around 70 hours to finish on 5 Tesla V100 GPU cards.

