# OpenReview forum: "The Remarkable Effectiveness of Combining Policy and Value Networks in A*-based Deep RL for AI Planning"
_ICLR.cc/2022/Conference — ICLR 2022 Submitted_

### Official Review · Reviewer_4bCR · 2021-10-25

**Correctness:** 2
**Technical Novelty And Significance:** 4
**Empirical Novelty And Significance:** 3
**Recommendation:** 6
**Confidence:** 4

**Main Review:**

## Strengths

* There are a lot of interesting ideas raised in this work.
* The topic areas are relevant to ICLR, although the paper may find a larger pool of interested readers in venues like AAAI, IJCAI, or ICAPS.
* The abstract model proposed to explain the heavy-tailed distributions observed when planning with A* + learned policy and heuristic is interesting and provides helpful intuition.
* The results in Table 2 support the conclusion that using the policy in the proposed way is a good idea. However, without the number of seeds/trials, it is difficult to assess whether these results are statistically significant.
* Similarly, the results in Figure 4 are compelling. I would like to see more results in these directions, for example, in different domains.
* I am not aware of prior work that uses uncertainty-aware NNs to learn policies/heuristics which are then used in combination with random restarts. I think this is a nice idea and could have general applicability.


## Weaknesses

* There are a large number of experimental details missing, so much so that it is difficult to interpret and evaluate the reported results.
   * None of the standard hyperparameters for neural network training and architecture are reported.
   * The way that the Sokoban states are encoded is not clear (there are many plausible possibilities).
   * I am guessing that the actions are just encoded as the four options “up”, “down”, “left”, “right”. (A different approach would be to use the ground PDDL operators as the actions, which has been done in prior work [1].)
   * The number of random seeds or trials does not seem to be reported. Was only one model trained?
   * I am guessing that the value networks are trained by minimizing MSE on the distance labels, and the policy network is also trained by minimizing a classification loss on the action labels, but there definitely needs to be more explicit discussion and details about both.
   * There are several other missing details -- overall, the paper needs a lot more information to reach the level where it would be possible to reproduce the results.
* A supplemental PDF and code would help clear up some of the confusions above, but none were submitted.
* The proposed model to explain left-hand heavy tails in section 4 is described informally. Formal definitions would help with clarity.
* Theorems 4.1 and 4.2 are stated without proof. It is also unclear what $\mathcal{A}$ is, in the context of the theorem statements. I am guessing it is a randomized algorithm that chooses a search node to expand with probabilities according to the model in Figure 2 (right), but that is a big leap, and the details are unclear.
* There is a big gap between the scope of the paper as implied by the introduction and the actual scope of the results. The introduction discusses PSPACE-hard problems in a very general way, but the results are ultimately concerned with a specific planning algorithm in one specific domain with a specific learned search guidance strategy.
* The references are missing connections to the learning for planning literature, which often appears in AAAI, IJCAI, or ICAPS. Some examples: [1-4]. Also, [5], but that one is not published. The connection to [1] is especially worth delving into. They propose a prioritization scheme that combines policy and value networks, similar to what is done here -- see their equation (22). It’s not exactly the same, so it would be interesting to explore the trade-offs.
* I would be curious to see baseline comparisons with the method proposed in [4]. Rather than altering the heuristic function, that method proposes to use a policy by running it for several steps during node expansion. Similar rollouts are used in [1] too. Based on the discussion of “routine macro actions”, it seems like these rollouts would do well here.
* There are a fair number of typos throughout. I noted some in the minor comments below.
* Overall, the paper would benefit from additional focus. The paper strikes me as trying to explore too many research threads at once, and it does not do justice to any of the individual threads. The threads are all very interesting, though.


[1] “Generalized Planning With Deep Reinforcement Learning.” Rivlin et al. (2020).
[2] “Neural Network Heuristics for Classical Planning: A Study of Hyperparameter Space.” Ferber et al. (2020).
[3] “Learning Domain-Independent Planning Heuristics with Hypergraph Networks.” Shen et al. (2020).
[4] “Learning Control Knowledge for Forward Search Planning.” Yoon et al. (2008)
[5] “Reinforcement Learning for Classical Planning: Viewing Heuristics as Dense Reward Generators.” Gehring et al (2021).



## Minor Points

* Typo: orders of magnitude *shorten* running time
* “The randomly generated instances lack sufficient structure and their underlying combinatorial search space is, in some sense, too regular.” The first part and the second part of this sentence seem to contradict each other.
* Grammatical typo: “the majority of instances have solution length contains hundreds or even thousands of moves”
* Typo: “The policy heuristics has”
* Typo: “in the training set even has been trained on”
* Typo: “an accuracy heuristic”
* The open quotation marks throughout the paper are usually facing the wrong direction
* The second column of table 2 is titled “A* cost function”, but it may be better to title that something like “priority function”, because it’s not really A* unless g(n) + h(n) is the priority function. These methods are variations on heuristic search more generally, not A* search specifically.
* Macro-actions are traditionally a sequence of actions that are executed open-loop. The “routine macro actions” described in section 5.1 sound more like a policy, since the actions will depend on the state.
* This sentence is imprecise, and I don’t know if I believe it: “It is widely believed and experimentally confirmed that with polynomially increasing model size, deep networks can achieve ”exponential” scaling power to unseen states.” What does “exponential scaling power” mean?
* It would be interesting to know what the states and actions at the critical nodes and the respective subtrees look like, qualitatively, in Sokoban. Are these “mistakes” similar to what a human would do in any way?

**Summary Of The Paper:**

This paper considers A* planning with learned policy and value function approximations. After presenting empirical results for heavy-tailed distributions of guided search runtimes, the authors propose an abstract model to explain these distributions. They also propose to search with random restarts as a way to address the heavy tails; and to perform random restarts effectively, they propose to use uncertainty-aware NNs, specifically ones with test-time dropout. Empirical results are in Sokoban.

**Summary Of The Review:**

This paper is somewhat extreme along multiple dimensions. On the one hand, there are many more interesting ideas than is typical for a single paper. On the other hand, there are many more experimental details missing than is typical even for rejected papers. The writing also needs to be more clear, precise and formal; the experiments and analysis need to go deeper; and the scope of the abstract and introduction needs to be narrowed. Overall, I think this could turn into a very good paper (or several papers), but the changes required will likely be too substantial to accept it for this conference.

---

> ### Author Response · Authors · 2021-11-19
> **Official Response to Reviewer 4bCR**
>
> Huge thanks to the reviewer. Please refer to the general reponse first.
>
> >There are a large number of experimental details missing......
>
> We apologize for the missing descriptions in our first version. We have added substantial quantitative data. New Tables 2 and 4, Figure 2, and more metrics to old tables were added. Specifically, Table 2 studies the performance comparison between the specialized solver and DNN-based search, Table 4 studies the more effectiveness of the policy over the value network, and Figure 2 is our new metric to separate left and right-hand heavy tails visually. Also, detailed descriptions of the network architecture and experiment details were added to Section 3 (Formal Framework) and the appendix.
>
> Specifically, the new Table 2 shows that given 10 mins, DNN-based search can solve 69% of hard instances that Sokolution cannot solve even given 2 hours. That is surprising. This really shows how the DNN-based search captures search guidance not found in the specialized Sokolution solver. Note also that in 10 mins, DNN-based search explores about 22,200 nodes, whereas, in 2 hours, Sokolution explores (unsuccessfully) about 4,449,600 nodes!
>
> >None of the standard hyperparameters for neural network training and architecture are reported. The way that the Sokoban states are encoded is not clear
>
> We have written Section 3. More details in Appendix.
>
> >The number of random seeds or trials does not seem to be reported. Was only one model trained?
>
> We use Monte Carlo dropout to generate various network checkpoints. Indeed, training multiple networks (ensembles) or using multiple snapshots of a single train can also serve as our purpose here, as shown in "A Survey of Uncertainty in Deep Neural Networks". We admit that to get low-variance results, multiple seeds and trials are needed. In the paper, experiment data about each instance has > 200 randomized runs, which makes the results of a single trial more robust.
>
> >A supplemental PDF and code would help clear up some of the confusions above, but none were submitted.
>
> We have rewritten Section 3 and the Appendix to contain all missing information. Sections 4 and 5 have also been substantially revised to better describe our observations and findings. The Sections also contain new data (Tables 2 and 4, and Figures 2). We will publish all codes if/when the paper proceeds to the next phase.
>
> >The proposed model to explain left-hand heavy tails in section 4 is described informally. Formal definitions would help with clarity.
>
> We have moved all definitions and results about left-hand heavy tails to Section 5 (Analysis of Left-hand Heavy Tails) in a self-contained way. The formal definition of left-hand heavy tails is now in Part "Abstract search tree model" of Section 5.
>
> >Theorems 4.1 and 4.2 are stated without proof. It is also unclear what \mathcal{A} is, in the context of the theorem statements. I am guessing it is a randomized algorithm that chooses a search node to expand with probabilities according to the model in Figure 2 (right), but that is a big leap, and the details are unclear.
>
> Proofs were added to the Appendix. The definition of \mathcal{A} is formally stated in Part "Optimal speedup of Las Vegas algorithms" of Section 2.
>
> >There is a big gap between the scope of the paper as implied by the introduction......
>
> We have narrowed down the scope in the introduction and changed the title (new title: “Left-hand Heavy Tails and the Effectiveness of  Policy and Value Networks in DNN-based best-first search for Sokoban Planning.”), and sharpened our contributions. See also below.
>
> Our experiments and experimentation are indeed on a single PSPACE complete domain, Sokoban planning. We have stated this more clearly in our revised writeup.
>
> The only domain knowledge of Sokoban our framework used is 1) state representation, 2) state-action transition function, and 3) deciding goals. No other Sokoban-specific techniques and heuristics are used. These three components are the *minimum* requirements to describe a planning domain. Also, we have tested on more than 10,000 instances with great variations in the underlying structure (see Paragraph "Data preparation" for more details). So, we believe the results will generalize to other planning domains.
>
> We focus our detailed experiments on the Sokoban planning domain because it has many instances beyond the reach of traditional AI planners (see above) and even specialized solvers such as Sokolution. Sokoban also has a uniquely large set of non-random instances to train on (about 10,000). Our DRL framework learns search guidance with little domain knowledge and little tuning. A core goal of ML enhanced approaches is to achieve superior performance in a given domain.
>
> >The references are missing connections to the learning for planning literature......
>
> All literature suggested by the reviewer was carefully surveyed and cited.

---

> > ### Comment · Reviewer_4bCR · 2021-11-29
> > **Thanks for the response**
> >
> > Thank you for addressing my feedback. The added details are helpful and the revisions look like an improvement (although I think the new title is too verbose). I will raise my score to a 6.

---

> > > ### Author Response · Authors · 2021-12-02
> > > **Thanks for the response**
> > >
> > > We thank the reviewer for the insightful review, feedback, and new score. We will try to come up with a shorter title that contains all key aspects of the paper. Thank you so much!

---

### Official Review · Reviewer_whsB · 2021-11-02

**Correctness:** 2
**Technical Novelty And Significance:** 3
**Empirical Novelty And Significance:** 2
**Recommendation:** 5
**Confidence:** 4

**Details Of Ethics Concerns:**

No concerns.

**Main Review:**

This paper empirically studies the behavior of the A* search when it is combined with the DNN heuristic generator (A*+DNN). This paper focuses on the Sokoban domain covering more than 10,000 instances available in the literature and presents some evidence that the running time distribution of A*+DNN may have left/right-heavy tails.

#### Strength of this paper
* large coverage of problem instances
* offering a connection with PHS (Orseau & Lelis 2021) and many earlier works in SAT/CP on heavy-tail distributions of the running times.
* intuitive explanations that may explain the results with the imbalanced tree model.

#### Weakness
* No information about DNN heuristic generator other than it used dropout 30%
* Lack of supporting experiments or claims for the 5 contributions author claimed.

#### General questions
  * This paper studies the Sokoban domain. How does the observation made in this domain applicable to AI planning in general?
  * How did you generate Figure 1? Did all 4 distributions come out from 1 Sokoban problem instance? Then, how did you process the running times to obtain the distribution?
  * What is the Monte-Carlo dropout mentioned in Figure 1?
  * Did you encounter memory limits during the experiments ? What fraction of the instances failed because of the memory and the rest because of the time limit?
  * Considering forward state space search in AI planning that commonly relies on best-first search with various heuristics, or memory-bounded best-first search, or searches with look-ahead, or hybrid type of search that mixes dept-first and best-first type search strategies,  this paper explores the basic A* with skipping the efforts ensuring the optimality of the solution. Have you compared the result with modern AI search algorithms other than PHS in Table 2?
  * What are the typical problem statistics? the length of plans, the size of the state space, etc?
  * Dropping the admissibility, A* is a more or less greedy search algorithm. In this paper, it is treated like that by comparing with variants of the heuristics in Table 2. Why did you include weighted A*? It is normally applied in the context of any-time search that offers a faster sub-optimal plan with the guarantee on the solution quality. Here, I don't see any good fit for using WA*.
  * The number of node expansions is very small (< 10^4) compared with the usual heuristic search (> 10^9). Is this because of the implementation limit? What was the time limit for the search? In this sense, the experiment may cover the cases when the search doesn't expand too many nodes.
 * What is the quality of heuristic estimates? How does a trajectory generated by DNN align well with the solution?
 * On page 8 under observation (1), did you see a case that supports your conjecture? "the distance heuristics are trained from solution paths so it possesses strong locality property".
  * On page 8 under observation (2), can you clarify "In theory, the depth term does not contain extra heuristic information about the search space", can you point out what theory supports that statement?
  * On page 9, can you provide references for the statement "it is widely believed and experimentally confirmed that with polynomially increasing model size, deep networks can achieve "exponential scaling power to unseen states".
  * Following the title of the paper, have you experimented with any other non-random AI planning domains?
  * Thm 4.1, and Thm 4.2, how to prove them?

* Comments on each contribution
1. For me, it reads like "the surprising effectiveness of the policy network" is an improvement of the coverage from PHS that replaces $\pi$ with $p$ in Table 2. Dividing the f with $\pi$ or $p$ gives more weights on the trajectories likely to be generated by DNN. On the one hand, using $p$ may improve the numerical stability. I think this choice enforces the search to follow the trajectory generated by DNN. Then, A* offers minor fixes on the trajectories generated by DNN. If DNN heuristics gets weaker, would you expect to see similar trends?
2. In the paper Figure 4. would be the only data that supports the empirical statement, which seems not sufficient. Do you have more results supporting this claim?
3. Probably the model explains the observation. But still, this paper only shows 1, which is not sufficient.
4. Can you provide the details on the network? I cannot find which part of the paper shows the importance of using uncertainty-aware networks.
5. Figure 4. b supports this claim. But I think we need more elaboration on the experiment. The time interval of restarts, some additional separation within the training, test, and the validation set based on the length or difficulty or the number of dead-ends in the problems.







**Summary Of The Paper:**

This paper empirically studies the behavior of the A* search algorithm with inadmissible neural network heuristics for solving Sokoban games.
* The paper introduces related works in SAT/CP on the "heavy-tail distribution" of running times and briefly summarizes the neural network set up for heuristic functions.
* This paper proposes an imbalanced tree model similar to the one shown in the context of SAT/CP and
makes some statements on the running time in Thm 1 and 2.
* The experiment section compares the coverage of the problem instances and statistics on the search space from randomly selected instances.

**Summary Of The Review:**

This paper shows interesting experiment results. However, we need clarification of the experiment setup and empirical results.
The idea is interesting and the approach is promising to understand the behavior of search with DNN. This is empirical work, so it's hard to make definite statements on the correctness. However, I think experiments don't support the claims well.

---

> ### Author Response · Authors · 2021-11-19
> **Official Response to Reviewer whsB**
>
> We thank the reviewer so much! Please refer to the general reponse first.
>
> >This paper studies the Sokoban domain. How does the observation made in this domain applicable to AI planning in general?
>
> The domain knowledge of Sokoban our framework used is 1) state representation; 2) state-action transition function; 3) deciding goals. No other Sokoban-specific techniques and heuristics are used. These three components are the *minimum* requirements to describe a planning domain. Also, we have tested on more than 10,000 instances with great variations in the underlying structure (see Paragraph "Data preparation" for more details).  So, we believe the results will generalize to other planning domains. However, we are stating our findings more carefully in our revised version. In particular, we emphasize that our experiments are on Sokoban planning instances, including very hard instances (e.g., Table 2). We take the Sokoban domain as a combinatorial challenge. The hard Sokoban instances are far out of reach of general AI planners and specialized solvers. Table 3 in the NeurIPS paper Feng el. shows general planners scale poorly on Sokoban.
>
> >How did you generate Figure 1?......
>
> Detailed caption about Figure 1 and description text followed was added.
>
> >What is the Monte-Carlo dropout mentioned in Figure 1?
>
> We have reorganized the paper so that Monte Carlo dropout is described in Section 3 before Figure 1.
>
> >Did you encounter memory limits during the experiments?
>
> The bottleneck of specialized Sokoban solvers is time. Festival (another specialized Sokoban solver) uses 16G memory at most and Sokolution uses less. Due to slow evaluation, neural network-based methods expand way fewer nodes so the memory usage is even smaller.
>
> >Considering forward state space search in AI planning that commonly relies on best-first search with various heuristics..... Have you compared the result with modern AI search algorithms other than PHS in Table 2?
>
> We added Table 2 to compare our DNN-based best-first search with Sokolution, which is the most effective (non-learning) specialized Sokoban solver. Sokolution uses AI-based search with many of the techniques you mentioned above. The Sokolution code represents many years of development.
>
> Our result shows that even with the simple best-first search framework, when augmented with DNN's heuristics, the algorithm can solve a significant number of hard instances that Sokolution cannot solve even given a 12x time limit. Notice that Sokolution out-scales general AI planners in the Sokoban domain.
>
> Specifically, the new Table 2 shows that given 10 mins, DNN-based search can solve 69% of hard instances that Sokolution cannot solve even given 2 hours. That is surprising. This really shows how the DNN-based search captures search guidance not found in the specialized Sokolution solver. Note also that in 10 mins, DNN-based search explores about 22,200 nodes, whereas, in 2 hours, Sokolution explores (unsuccessfully) about 4,449,600 nodes!
>
> >What are the typical problem statistics?......
>
> We added all these statistics in Part "Data preparation" of Section 3 (Formal Framework).
>
> >Dropping the admissibility......Here, I don't see any good fit for using WA\*.
>
> Our initial motivation was to test whether WA\* can solve more instances than A\* in trade of plan quality. Interestingly, as shown in Table 3, A\* can actually solve more instances than WA\* under most search budgets. This motivated our further experiment to study whether the learned value heuristic is ill-informed (see Table 4).
>
> >The number of node expansions is very small (< 10^4) compared with the usual heuristic search (> 10^9)......
>
> In addition to node expansions, we added the runtime metric to all experiments. Value and policy networks are relatively slow to evaluate so the expanded nodes are usually orders of magnitude smaller (about a factor of 15). However, the learned networks provide far superior search guidance as shown in the new Table 2. DNN-based search with small node expansion is still competitive and even outperforms specialized solvers on hard instances (solving 69% of hard instances v.s. 0% for the specialized Sokolution solver).
>
> >What is the quality of heuristic estimates? How does a trajectory generated by DNN align well with the solution?
>
> Figure 4 gives a good visualization of the alignment.
>
> >For me, it reads ...... is an improvement of the coverage from PHS that replaces......
>
> Our main message is not to show our evaluation function f(n) is better than PHS. Indeed, we show the learned policy heuristic is significantly more effective than the value heuristic. Because in traditional best-first search, the evaluation function f(n) only consists of g(n) and the value heuristic h(n). Machine learning enables learning an extra policy term p(n). It is interesting to study the impact of this new term on traditional best-first search.

---

### Official Review · Reviewer_VVfA · 2021-11-04

**Correctness:** 2
**Technical Novelty And Significance:** 2
**Empirical Novelty And Significance:** 1
**Recommendation:** 1
**Confidence:** 5

**Main Review:**

# Thank you

Thank you for your comments. I see the new manuscript clarified many points.
Unfortunately, I cannot honestly change my general opinion about the acceptance of the paper.
The Sokoban part per see is interesting, although not enough for acceptance.
The notion of left-heavy tail is something else. In figure 3-right, the problem can be just very hard to solve.
For many algorithms, there are classes of problems that generate a worst-case behaviour.
I wonder if I could be deeply confused, and tried to challenge my understanding given the comments made and the improvement.
I couldn't. If you insist this is important, please re-submit to a search-minded venue like Socs,
or to a journal like the AIJ or JAIR.

# Previous main review
Before examining the contributions, I feel obliged to clarify some points that I'd use in the rest of the review.

1. The abstract does not mention that this empirical work is for one domain in particular: Sokoban. Two theorems are stated in the paper but without proofs. That means the observations must be circumscribed to the experiments that compare A* using
	- A very basic heuristic
	- PHS: DRL policy from previous work
	- PHS*: a variation of PHS proposed here.
2. That leaves only two interesting cases in the experiments: PHS and PHS*. So, the observation about the distribution of complexity and the conjectures are circumscribed to Sokoban and these two related heuristics.
3. The paper refers to heavy-trail distribution in SAT solving. That's an interesting connection, but it's important to consider that SAT solvers are general tools, not specialized algorithms for specific domains.
4. Moreover, the paper seems to say that A* is a planning algorithm. The culture of the research community always mediates the name of techniques. Now, the search community –that publishes in SOCS, ICAPS, AAAI and IJCAI– would say that A* is a search algorithm that could be used for planning.
	- For instance, specialized algorithms for Sokoban use search algorithms like A*. The search community won't call them planners.
	- However, some DRL papers say that MCTS is a planning algorithm, while MCTS is a search algorithm like A*. I do respect that different communities might use different names.
	- For completing the picture: the planners that participate in the International planning competitions (IPC) are general solvers, like SAT solvers are. The simple class of problems in the IPC is PSPACE-complete. However, it doesn't mean that all the specific problems are PSPACE-complete. That leads me to the next point.
5. The observations about the distribution depend on the distribution of the original problem. As shown in many combinatorial domains, algorithms tend to have strengths and weaknesses depending on the domain. For instance, in the SAT competitions, the winner of the industrial track tends to be different from the random track.
6. That means that observations on the distribution of behaviour need to be grounded on the problem distribution and might not translate in the same way to other algorithms.

These six points above justify the summary I made about this paper: the paper is not about A* + DRL for policy/value. Instead, it's about Sokoban and two specific DRL techniques that end up having properties previously observed in SAT solving, and domain-independent planning as SAT.

Moreover, the comparison between using A* with DRL heuristics and other methods is not trivial because of the following.

Typically, non-ML based specialized search algorithms have very efficient node generation. For instance, the reference below discusses cases of over 1 million nodes expanded. On the other hand, DRL networks tend to have lower generation speed, but the training time is not accounted for and can provide an edge in certain instances. That means that neither of these metrics allow a complete picture of the comparison between non-ML vs NN-based heuristics:
- Running time
- Number of expansion.
So, in table 2, the no-policy versions might generate nodes much faster than the policy-based. On the contrary, it'd make sense to compare node expansions of PHS and PHS+.

I'll finish this section of the review with some questions I'd like to answer in the rebuttal. Then I'll continue with some more comments about the paper.

## Questions to be answered in rebuttal.

0. Could some properties of the instance explain the left-tail? For instance, they could be mostly small instances or very easy to solve.
1. How many nodes per second expands A* for the different configurations mentioned in Table 2?
2. The use of SPFA reported before section 4 make it hard to compare the number of expanded nodes with previous work. It is also hard to compare the findings with previous work on A*. How do the experimental results look like if SPFA is not used?
3. Page 2 says: *In fact, deep learning guided search algorithms that combine traditional search-based methods such as A star with deep neural networks heuristic prediction have shown promising progress. These methods can solve a significant number of hard planning instances that specialized solvers cannot solve.*
	- Please add concrete references and comment on the comparison.
	- Please take into account that specialized solvers tend to assume a higher running time, and that the good performance might depend on properties of the problem like size. For instance, A* + nontrivial non-ML heuristics might perform well in big environments where the agent needs to travel long distances to move between a few boxes.
4. What's the heuristics PureDistance?
	- Were other heuristics considered? Indeed, specialized solvers use heuristics too.
5. How does  Sokosolution compares with the rest of the experiments? Please report running time, expanded nodes, and nodes per second.
6. How costly is to train the DRL heuristics?
	- What parameters were used?
	- How were the hyper-parameters tuned?

## Additional comments

page 1: *Recent work based on deep learning guided search algorithms that combine traditional search-based methods, such as A and MCTS search, with deep neural networks' heuristic prediction has shown promising progress*.
Please add a reference. There is work all the way from TSP to game-playing. I'm not sure what previous work the paper should consider as relevant.

#### Introduction

page 1: *such as problems captured by the Planning Domain Definition Language (PDDL) (Bylander, 1994) and Sokoban*
Confusing. I'd switch Sokoban and PDDL. By the way, Sokoban can be expressed in PDDL. Sometimes that's used as a homework: https://github.com/Tarrasch/sokoban-planner/blob/master/sokoban-pddls/sokoban-domain.pddl

page 1: *(QBF), the canonical example of a PSPACE-complete problem*.
Yes. A reference, please

page 2: *These advances naturally raise the question of whether the success in game domains can be transferred to PSPACE-hard planning domains.*
As I mentioned above, trained DRL for game-playing can play one kind of game. So, trained DRL can be compared with a specialized solver. However, suppose the same architecture can be trained in for multiple environments. In that case, they are more general than specialized solvers, depending on how hard they are to tune and the sample complexity. The main question is always how hard it is to generate a good algorithm in a new problem.

page 2: *we explore and generalize the cost distribution profiles of search methods from NPhard domains to PSPACE-hard planning domains, and show heavy-tailed cost distributions exist ubiquitously among planning instances.*
and page 2: *We identify heavy-tailed runtime distributions of PSPACE-hard planning problems*
This holds for Sokoban with these particular heuristics.

page: *Such heavy-tailed distributions have been observed before in NP-hard domains, such as SAT and CSP (Gomes et al., 1997; 1998) but not in PSPACE settings.*
Not true. See Eldan Cohen, J. Christopher Beck:
Fat- and Heavy-Tailed Behavior in Satisficing Planning. AAAI 2018: 6136-6143.
Satisfying planning of classical planning is PSPACE. That includes Sokoban.

page 2: *We study the interplay of the policy and value networks in A*-based deep RL. Our experiments show the surprising effectiveness of the policy network*
Why is this surprising?

page 3: *We show how a restart strategy can improve the deep RL planner effectiveness. In particular, our experiments show how for a given search budget, there is an optimal restart strategy. For larger budgets, more frequent restarts are most effective.*
This depends on the algorithm. State of the art planners sometimes use variations of A* that use extra information from the domain or follow the heuristic more greedily. Search for "greedy best first search" and "enforced hill-climbing". See also the planner FastDownward: http://fast-downward.org.

#### Background and Related Work

page 3: *Luby et al. (1993) prove that when we have full knowledge about the distribution p A, the optimal strategy that achieves the minimum expected time*
What about hard tasks that need more time than t_A?

page 3: *figure 1 shows the comparison of heavy tails and exponential tails on various statistics.*
Is this A* over Sokoban or a synthetic problem such that t_A can be defined analytically?

page 4: *Orseau & Lelis (2021) add path probability π(n) to the minimizing term of A star search*
What's the minimizing term of A*? It searches following the min f=g+h, where g is the cost and h is the heuristic value. Here \pi(n) is a probability. How is that added?
I see this explained in more detailed below. I suggest to say here that "O&L divide the minimize value by \pi(n)" or weigth by 1/\pi(n).

page 4: *The search quickly degenerates into BFS when goals require a long plan since shallow nodes are always preferred by A star based on π(n). Our method modifies the term π(n) to p(s l|s l−1) to avoid exponential growth of the minimizing term of A* as the depth of nodes grows.*
It depends on the heuristic used. Is \pi(n) as used a heuristic? An heuristic is an estimate of the cost of achieving the goal. Both terms of \pi(n) would produce values < 1, while the usual cost in Sokoban is 1 per action.

#### Data preparation

page 5: *remaining unsolved 2609 instances as the hard set to further study the cost distribution of instances that are way harder than the training instances.*
They could be slightly harder as it'd take 11 minutes to solve them. I suggest to sample a subset of them and try to solve in 30m or 2h.

page 5: *move" consists of four actions: upward, downward, rightward, and downward. The framework only needs to detect whether the four adjacent cells are empty, or whether the box is pushable if one occupies an adjacent cell.*
Another used alternative is to define push a requiring an adjacent box.

#### Policy-guided A* search

page 5: *We use the path length as the cost of A* search so that g(n) is the depth of node n.*
I guess the implementation detects repeated nodes. Please clarify

page 5]: PHS star is a significantly improved version*
Say before that the change is called PHS*

page 5: *We found the combinatorial search is so challenging that the model cannot reliably solve planning instances in the training set even has been trained on ground truth labels from it.*
I think this argument does not follow. The negative empirical result does not imply that the problem is more or less challenging. I'd accept a weaker statement of the form "The model couldn't solve instances used for training".

page 6: *As the evaluation time of deep networks heavily depends on hardware, we use the number of expanded nodes of A star as the runtime cost instead of the actual running time.*
This is standard in search research

page 6: *We consider duplicate board detection so that various nodes with the same board will merge into a single node*
Say this early to make sense of the cost being the path.
Minor question: what if a shortest path if found later? Optimal A* reopens visited nodes/states, unless properties of the heuristic imply that's not necessary. In this case, optimality is not required. However, it'd be better to say it clear:
- duplicate states are not expanded.
- The cost of a node is the cost of the 1s path to it.

page 6: *To increase the algorithm's efficiency and improve the evaluation metric, each time a node n is popped out from the open set (frontier nodes of the search), we do a local Shortest Path Faster Algorithm (SPFA) starting from n to ensure that no relaxation exists in the closed set at any stage of A star search, i.e., g(u) + 1 ≥ g(v) for each pair of adjacent nodes u, v while u is in the closed set.*
This is non standard. I hope an ablation study is done about the impact of this. This description is not enough for reproducing the results


#### Heavy tails on the left

page 6: *Our proposed model assumes the existence of O(log(n)) critical nodes among the plan from which a wrong child node selected by A star will result in extra exponential search space.*
Why log(n)? What's N in this case? For NP problems, the upper bound is O(n) because that's exactly the complexity in a non-deterministic Turing machine.
If this a conjecture, please say so. The conjecture would have value if the empirical results validate it. However, Sokoban is PSpace-complete, so I don't expect this conjecture to be validated in the Experiments section.

page 6: *Theorem 4.1*
Proof of both theorems are not in the main body of the paper. There is no appendix

page 7: *We empirically found that in the search graph of A, the majority of nodes do not branch — heuristics provided by the network is accurate enough to prefer the right child node with high confidence.*
This general statement is at odds with the idea of heavy right tail. It'd be inconvenient to change this statement to talk about the mean since that's not well defined for heavy-tail distribution.

page 7: *To explain why SAT solvers work so well in practical instances, Hoffmann et al. (2006) examine various benchmarks and identify that for most practically solvable SAT instances, after assigning values to logarithmic variables, the remaining problem instance quickly becomes polynomially solvable by propagation rules.*
Hoffman et al examine planning instances solve by translation to SAT. Those propositional theory have a particular structures that might not apply directly to SAT. The backdoor conjecture was examined in previous work by the co-authors the Hoffman of that paper.

page 7: *This result illuminates the prototypical patterns of the structure causing the empirical behavior observed in the International Planning Competitions benchmarks.*
Reference? In general, it'd be good to discuss other domains of the planning competition. Some of them have complexity analysis like the referred paper on Sokoban being PSpace-complete.

#### Structure of real search graphs
page 7: *Whether AI planning systems can find routine macro action, i.e., a sequence of algorithmic actions to perform a sub-goal, has a great interest for researchers.*
This phenomenon is fairly common in instances that are solved using A*, when the solution is found quickly. The paragraph after this sentences implies this phenomenon is specific to this experiment. Instead, the paper show report experiments using A and a non-DL heuristic to see if the same phenomena appear. The non-DL heuristic could be a specialized solver or a general automated planner, even if it scales to smaller instances.

#### Effectiveness of policy and value networks

page 8: *5.2 EFFECTIVENESS OF POLICY AND VALUE NETWORKS*
What is Pure Distance? The effectiveness of A* depends strongly on the heuristic. If the heuristic is ill-informed, using WA* tends to degrade the performance.

Geometric Mean? Why? It's hard to qualify the meaning of this result without comparison with other algorithms.

The experiments are in a region of a relatively low number of expansions. See for instance, Pereira et al below where the number of expansions goes all the way to 1 million nodes.

Moreover, the conclusion about what's the best algorithm depends on how much time is available. The last full international planning competition (2018) included these tracks:
- satisfying. Score if instance solved in less than 30m.
- agile. Positive score proportional to speed until a limit of 5m.
https://ipc2018-classical.bitbucket.io

Even for the same algorithms, the best configurations is different depending on the speed required.

André G. Pereira, Marcus Ritt, Luciana S. Buriol,
Optimal Sokoban solving using pattern databases with specific domain knowledge,
Artificial Intelligence,
Volume 227,
2015,
Pages 52-70,
ISSN 0004-3702,
https://doi.org/10.1016/j.artint.2015.05.011.

page 8: *(3) Weighted A start does not help.*
This observation was done in a relative low number of expanded node.

#### Experiment data for the abstract model

page 9: *Hoffmann et al. (2006) has given an explanation to STRIPS-style planning with backdoor models — after finding and assigning logarithmic variables the remaining problem solving becomes polynomial. In contrast, finding these critical nodes is not necessary for expand-style search algorithms*
This is for SATPLAN: classical planning as SAT

page 9: *It is widely believed and experimentally confirmed that with polynomially increasing model size, deep networks can achieve" exponential" scaling power to unseen states*
Widely believed by whom? I don't believe that myself as a general statement. It is an interesting hypothesis, but it needs to be studied. A source of confusion is the tension between solving single problems vs general solvers. It does make sense that some learning based algorithms might be more powerful than specialized solvers. But what worked in one domain might not translate into another one.

End of the day, what matters is the resources necessary  to get the solutions. If the same domain is to be solved multiple times, like Go, training makes sense. On the other hand, if many variations are solved, it'd make more sense to have a general solver.

#### References

page 10: *Joseph Culberson. Sokoban is pspace-complete. 1997.*
Check the font case of the references. Pspace, sat and others are upcase. Rubin is a proper name.

page 11: *Laurent Orseau and Levi HS Lelis. Policy-guided heuristic search with guarantees. arXiv preprint preprint arXiv:1802.10501, 2018. arXiv:2103.11505, 2021.*
This was accepted to AAAI 2021. Please avoid arXiv references when the paper has been accepted.

Finally, here are two interesting references:
- Yaron Shoham and Gal Elidan. "Solving Sokoban with forward-backward reinforcement learning". SOCS 2021. (Both in arXiv and in the socs website, where this is a video too).
	- This might be a good algorithm for comparison.
	- I still think that's fairly standard in search. In a sense, the left-tail was always there as.
- This is an undergrad thesis. I cite it with the best of intentions. It showed up as I was looking for references. It discusses cases where Sokoban becomes harder for non-specialized planners (using PDDL).
	- https://ai.dmi.unibas.ch/papers/theses/haenger-bachelor-13.pdf
- This is another relevant undergrad thesis
	- https://baldur.iti.kit.edu/theses/SokobanPortfolio.pdf

See below some extra problems that are also PSPACE-complete:
- PSPACE-Completeness of Sliding-Block Puzzles and Other Problems through the Nondeterministic Constraint Logic Model of Computation. Robert A. Hearn, Erik D. Demaine
- Aviezri S. Fraenkel and Elisheva Goldschmidt. 1987. PSPACE-hardness of some combinatorial games. J. Comb. Theory Ser. A 46, 1 (Sept. 1987), 21–38. DOI:https://doi.org/10.1016/0097-3165(87)90074-4


**Summary Of The Paper:**

The paper study the behaviour of A* combined with Deep-RL (DRL) based heuristics in Sokoban, a PSPACE-complete problem, and an important domain in search, planning and RL. Experiments show a heavy-tail distribution on running time in some cases; that imply abundant cases where the problems are solved quickly –in polynomial time– or take a long time. Furthermore, the paper shows how previous search ideas can help solve more problems quickly: restarts and randomization.

The paper enumerates five contributions, but I am not convinced of their value. Instead of presenting them and then commenting about them, I will go over them and comment. That would lead me to a conclusion about the paper. I will recommend rejecting the submission as this needs further work.

Despite my interest in this line of work, my recommendation is studying the interaction between search algorithms and ML-based heuristics, including DRL.

**Summary Of The Review:**

In summary, I think the current manuscript extrapolates empirical observations in one domain -Sokoban– with a few related heuristics. For showing something specific about them, other heuristics or specialized solvers should be used. I wouldn't be surprised if different algorithms solve fast different kind of problems.

I understand the intention of the contribution is about DRL-based heuristics exploiting the so-called left-tail. However, I don't want to made mistake I was pointing out. The question is whether that observation is a property of the DRL proposed here, or a fairly common situation that appears in other cases. Comparing with specialized solvers and other ML-based heuristics might help to answer this critical question.

Therefore, I recommend to reject the paper for ICLR 2022.

I think this paper would be a more appropriate for SOCS, ICAPS or a general conference like AAAI, IJCAI or ECAI.

For the sake of clarity, I'll react to the contributions:
1. We study the interplay of the policy and value networks in A*-based deep RL. Our experiments show the surprising effectiveness of the policy network, further enhanced by the value network, as a guiding heuristic for A*.
	- This observation could be significant for solving Sokoban, but it's not.
	- This observation could be significant for DRL-based heuristics, but it's not.
	- Both possibilities would require comparison with other algorithms, including specialized solvers, unless other problems beyond Sokoban were considered.
2. We identify heavy-tailed runtime distributions of PSPACE-hard planning problems and propose a series of distribution-independent statistics to quantify the heaviness of tails and effectiveness of random restart. For the first time, we show and extensively study heavy left tails from experiment data, introduce an abstract tree search model with critical nodes, and formally show how heavy left tails can arise during the search.
	- The so-called left-tail phenomena appears in many techniques using search. See other papers about Sokoban mentioned here. The amount of instances tested could be smaller, but the high diversity of running time is consistent with the findings here.
	- In any case, given the experiments, this observation would be valid only for Sokoban, and the reader must wonder if there is something special on the DRL-based heuristics.
3. The tails on the left of the runtime distribution are explained by the critical role of the policy network as a guiding heuristic. Polynomial runtime scaling can occur because the policy network helps avoid exploring exponential size sub-trees early on in the search. figure 3 visualizes the phenomena by revealing a small set of critical nodes in the early part of the search space.
	- This is also fairly common in search. I don't think this is a contribution. The same sentences "Polynomial..." Apply to the behaviour of a SAT solver.
4. We show the importance of using uncertainty aware networks in the planning domain and how it can add a controllable amount of randomness to a backtrack-style solver.
	- This holds. The difference is of 4% absolute.
5. We show how a restart strategy can improve the deep RL planner effectiveness. In particular, our experiments show how for a given search budget, there is an optimal restart strategy. For larger budgets, more frequent restarts are most effective.
	- This is an interesting observation. It's possible that DRL-based heuristics benefit more from restart in comparison with specialized solvers and generic domain-independent planners

---

> ### Author Response · Authors · 2021-11-19
> **Official Response to Reviewer VVfA**
>
> We thank the reviewer so much! Please refer to the general reponse first.
>
> In the revision, we avoid using the term "general solver". Instead, we refer to DRL as a general framework, as done in the literature, because the framework only needs 1) state representation; 2) state-action transition function; 3) deciding goals, to learn one planning domain from scratch. No other specific domain information is used. So, our combinatorial search and learning framework is quite general. Nevertheless, we are careful to point out that our experiments focus on the Sokoban planning domain (See also our new title!). Sokoban is one of the hardest-known domains for AI planners. Table 3 in the NeurIPS paper Feng el. shows general planners scale poorly on Sokoban.
>
> In this paper, we do not use any extra knowledge about Sokoban besides the plans found by Sokolution. Also, we have tested on more than 10,000 Sokoban instances with significant variations in the underlying structure from different sources (instead of algorithmically randomly generated instances). So we believe our analysis results can generalize. We have added the main message of this paragraph in Section Introduction.
>
> >The so-called left-tail phenomena appears in many techniques using search.
>
> We realize the reviewer might equate left-hand heavy tails as the same as heavy tails as reported in the literature. So in Section 1 we clarify and highlight the difference. We believe this paper is the first paper to characterize and formalize left-hand heavy tails. We reached out to the authors who first showed right-hand heavy-tails in search (e.g., combinatorial problems, scheduling, and proof planning) who had also conjectured the existence of heavy-tails on the left-hand side for instances with exponential median time. They confirmed they had never empirically confirmed the existence of left-hand heavy tails and they were not aware of any other work showing such left-hand heavy tails.
>
> We would appreciate it if you could point us to a reference to left-hand heavy tails if such work exists. Note that the reference you provided (a relevant paper indeed) concerns right-hand heavy-tails in planning domains.
>
> >They could be slightly harder as it'd take 11 minutes to solve them......
>
> We extended the time limit to 2 hours to generate the hard instances.
>
> >Why is this surprising?
>
> The new table 2 shows that given 10 mins, DNN-based search can solve 69% of hard instances that Sokolution cannot solve even given 2 hours. That is surprising. Note that in 10 mins, DNN-based search explores about 22,200 nodes, whereas, in 2 hours, Sokolution explores (unsuccessfully) about 4,449,600 nodes!
>
> >......the paper is not about A\* + DRL for policy/value. Instead, it's about Sokoban and two specific DRL techniques......
>
> We focus our detailed experiments on the Sokoban planning domain because it has many instances beyond the reach of traditional AI planners (see above) and even specialized solvers such as Sokolution. Sokoban also has a uniquely large set of non-random instances to train on (about 10,000). Our DRL framework learns search guidance with little domain knowledge and little tuning. A core goal of ML enhanced approaches is to achieve superior performance in a given domain.
>
> Sokoban indeed represents a specific hard combinatorial search domain with training data. We conjecture that these results will also apply to other combinatorial search problems using the DRL + A\* framework, simply because it would be surprising if they only occurred on Sokoban problems. ***Nevertheless***, we have been more careful in stating our findings. See the revised text. Finally, we note (1) left-hand heavy tails have been conjectured to exist for SAT but not observed (to our knowledge), and (2) right-hand heavy tails were first observed and reported for SAT on a single combinatorial task, quasigroup completion (underconstrained instances). Later they were confirmed in many other domains.
>
> So, we stress again that our learning framework is general. However, part of the strength of the framework is that the learned search guidance is domain-specific. That is, the value and policy network learns how to solve hard Sokoban instances (by training on easier instances!! See Table 2).
>
> Such domain-specific heuristics (automatically learned) may well be the key to finding the left-hand heavy tails. A left-hand tail requires finding short successful searches on very hard instances. Having finely tuned learned search guidance may be necessary for such runs to be found. In fact, the surprising effectiveness of the Policy network suggests that the deep Policy net learns structure is not easily replicated in a domain-independent heuristic (Tables 2, 3, and 4).
>
> It's an interesting open question whether left-hand heavy tails will be found for SAT solvers. ***The general heuristics in the SAT solvers may not be strong enough to get the needed short runs on very hard instances***.

---

### Official Review · Reviewer_Lduu · 2021-11-07

**Correctness:** 2
**Technical Novelty And Significance:** 2
**Empirical Novelty And Significance:** 2
**Recommendation:** 3
**Confidence:** 4

**Main Review:**

Strength:
	1. This paper created an abstract model which explains the right- and left- heavy tail issues for best-first search algorithms.
	2. Experiments seem to demonstrate the effectiveness of the proposed combination of policy, valuation, random-restart and uncertainty networks with no or small modification to the original proposal in the literature.

Weakness
	1. No explicit description of the specific architectures of value network, policy networks, uncertainty aware networks used in this paper. This makes it difficult for the readers to replicate and evaluate the applicability of the approach used in this paper.
	2. The writing of the abstract model of heavy right- and left- tail problems need improvement to make it clear and mathematically solid. The notations and concepts need to be unified, be introduced formally and be referred in any usage. Also the abstract model seems to be a standalone contribution to the best-first search framework; and it doesn't need or make use of any properties of the deep nets heuristics or policy networks. Please clarify.


Details

   1. Luby et al. (1993) seems to be an important component of your abstract model of understanding the left- and right- tail issues. Please considering moving the math definition to section 4. Also the notation of p in section 4.1 seems to be conflicting with section 4.1 and a few other places. Please use and define notation explicitly and carefully.
	2. Page 4, a typo? \pai(n) = … p(s_l | s_l-1) , should l be n?
	3. Page 4, please definite critical nodes mathematically.





**Summary Of The Paper:**

This paper studied A* style best-first search with deep networks guided value and policy functions. Theoretical framework was created for the analysis of heavy left- and right- tails problem in running time. This paper also claimed that according to the experiments policy network, random restarts, and uncertainty aware networks are import contributors of solving hard planning problems.  However, this paper did not provide its own version of the policy, valuation and uncertainty networks making it difficult to evaluate the validity of the approach and the theoretical analysis --- I need to guess from the literature which exact networks are used in the experiments and the theoretical analysis.


**Summary Of The Review:**

This paper needs significant improvement in the writing clarity, mathematical soundness, and experimental validity to reach publication status. For example, this paper did not provide its own version of the policy, valuation and uncertainty networks making it difficult to evaluate the validity of the approach and the theoretical analysis --- It requires the readers guessing and jumping across the literature to get the important components neural net constructs, e.g. the valuation, policy and uncertainty aware networks.

---

> ### Author Response · Authors · 2021-11-19
> **Official Response to Reviewer Lduu**
>
> We thank the reviewer so much! Please refer to the general reponse first.
>
> > No explicit description of the specific architectures of value network......
>
> We apologize for the missing descriptions in our first version. We added detailed descriptions in the appendix.
>
> >The writing of the abstract model of heavy right- and left- tail problems need improvement......
>
> We moved all descriptions and results about left-hand heavy tails, including the abstract search tree model, to Section 5 (Analysis of Left-hand Heavy Tails) in a self-contained manner. The tree model applies to all best-first search algorithms based on heuristics with randomness, e.g., Monte Carlo dropout deep networks and traditional manually crafted heuristic portfolios. However, instead of exponential tails, heavy tails can only occur when the heuristics are powerful enough to avoid exponential space early during the search. We have clarified this in the revision.
>
> >Luby et al. (1993) seems to be an important component......
>
> We unified the math symbols in the revision. We also added detailed descriptions about our distribution-independent statistic methods in Section 5. Section 5 is now self-contained (in the section, the definition of critical nodes is described before the first reference).
>
> >This paper needs significant improvement in the writing clarity, mathematical soundness......
>
> Sections 4 and 5 are rewritten to avoid excessive subjective conclusions. We rewrote Part “Effectiveness of policy and value networks” to provide our own version of policy and value networks in Section 4. In Part “Solving more instances with random restarts” of Section 5, Figure 5b demonstrates the importance of introducing randomization with uncertain-aware networks. Specifically, given larger search budgets, more frequent restarts become more effective.
>
> Specifically, we added Table 2 to compare our DNN-based best-first search with Sokolution, which is the most effective (non-learning) specialized Sokoban solver. Sokolution uses AI-based search with many of the techniques you mentioned above. The Sokolution code represents many years of development.
>
> Our result shows that even with the simple best-first search framework when augmented with DNN's heuristics, the algorithm can solve a significant number of hard instances that Sokolution cannot solve even given a 12x time limit. Notice that Sokolution out-scales general AI planners in the Sokoban domain.
>
> Specifically, the new Table 2 shows that given 10 mins, DNN-based search can solve 69% of hard instances that Sokolution cannot solve even given 2 hours. That is surprising. This really shows how the DNN-based search captures search guidance not found in the specialized Sokolution solver. Note also that in 10 mins, DNN-based search explores about 22,200 nodes, whereas, in 2 hours, Sokolution explores (unsuccessfully) about 4,449,600 nodes!

---

### Author Response · Authors · 2021-11-19
**General Response to All Reviewers**

We thank reviewers' detailed comments and insightful questions which helped us improve the paper significantly. In this response, we will address the general changes we made to the paper. We will answer specific questions in individual responses. We sincerely invite the reviewer to go through the whole revised paper since we reorganized contents, added extra experiments, and corrected mistakes to the revision thanks to the reviewer's kind comments. We also list the main changes below to help the reviewer to notice the improvements.

1. We realized that our writing and exposition were unclear. We, therefore, have extensively rewritten and reorganized the paper. We have sharpened the statement of our three main contributions, each quantitatively supported: (1) An evaluation of the Policy and Value Networks showing the remarkable effectiveness of the Policy Network. (2) Identification of Heavy tails on the left, i.e., short successful runs on very hard instances + novel formal model. (3) Introduced the use of uncertainty-aware networks to obtain effective restart strategies. For Contributions 1, see Section 4, and for Contributions 2 and 3, see Section 5. The Appendix gives further details.
2. We apologize for the missing descriptions in our first version. We have added substantial quantitative data. New Tables 2 and 4, Figure 2, and more metrics to old tables were added. Specifically, Table 2 studies the performance comparison between the specialized solver and DNN-based search, Table 4 studies the more effectiveness of the policy over the value network, and Figure 2 is our new metric to separate left and right-hand heavy tails visually. Also, detailed descriptions of the network architecture and experiment details were added to Section 3 (Formal Framework) and the appendix.
2. We added proofs of Theorem 1 and 2, as well as the details about the network architecture, experiment settings, and the description of MC dropout (uncertainty) layers in the appendix.
3. We reorganized the paper and moved all new findings and results about left-hand heavy tails to Section 5 (Analysis of Left-hand Heavy Tails). We realized the reviewer might equate left-hand heavy tails as the same as heavy tails in the literature. So in Section 1 (Introduction), we clarify and highlight the difference.
4. A new metric (Figure 2) is added to Section 5 to visualize the difference between left and right-hand heavy tails. We further added the detailed descriptions about the sample statistic method we used to evaluate tails to Section 5.
5. New texts were added to the Introduction to discuss the generality of our framework.
6. We revised the title to better reflect our contributions by adding “left-hand heavy tails”, replacing “A\*” with “best-first search” and “AI Planning” with “Sokoban Planning”, and removing “remarkable” and “combining” to shorten it. New title: *Left-hand Heavy Tails and the Effectiveness of  Policy and Value Networks in DNN-based best-first search for Sokoban Planning*.
7. We improved the overall writing of the paper substantially. Significant details were added, especially in Section 3 (Formal Framework) and Section 4 (The Interplay of the Policy and Value Networks). In Section 4, we added a statistical comparison on node expansion, time, and the solving ratio between DNN-based search with domain specialized solvers. Because neural heuristics are slow to evaluate, in addition to node expansion, we added the runtime metric to all experiments.
8. We added a new experiment to Section 4 to compare the effectiveness between the policy and value network. See new Table 4.
9. We use words more carefully, e.g., from “general solver” to “the Sokoban domain” and from
"assume"/"believe" to "hypothesize" when the result is empirical. We added words to highlight that our analysis uses little domain knowledge of Sokoban.
2. We standardized terminology usage, e.g., by changing "A\*" to "best-first search", "PureDistance" to "greedy BFS", "planners" to "search algorithm", etc. We removed SPFA for standardization.
3. As suggested, we added experiment metrics about nodes expansion, solving time, nodes per sec, and the solved ratio between ML-based and non-ML-based methods.
4. We added more citations and avoided using Arxiv versions as suggested. Also, changed, polished, and standardized multiple word usages.
5. We read all the suggested missing references and corrected some mistakes (again, a huge thank you to the reviewer!), e.g., the misstatement that heavy tails had not been studied in planning. We added the suggested papers to the references (we also added a paper on heavy-tails in proof planning). Again we stress that our main point is about left-hand heavy tails, which we believe had not been studied before.
6. We rewrote Section 5 to better explain left-hand heavy tails in a self-contained way.

---

### Decision · Program_Chairs · 2022-01-20

**Decision:**

Reject

**Comment:**

The paper proposes a mechanism for A* planning with learned policy and value functions. The experiments (restricted to the Sokoban domain) show that the runtime of guided search follows a heavy-tailed distribution, suggesting that in many cases, the problem is either solved quickly or takes a long time. An abstract model is proposed to explain this distribution, and a number of mechanisms are proposed to overcome its challenges.

The reviewers thought the paper had some interesting ideas but found the experimental section to be especially weak. While the paper starts out with quite general claims, the experiments only consider a single domain. Also, key details about the experiments were missing. Finally, the writing feels rushed -- the original submission had many typos and lacked proofs for two theorems.

I agree with these objections and recommend rejection. Please revise the paper following the reviews and resubmit to a different deadline.